# Sevoflurane and Desflurane Exposure Enhanced Cell Proliferation and Migration in Ovarian Cancer Cells via miR-210 and miR-138 Downregulation

**DOI:** 10.3390/ijms22041826

**Published:** 2021-02-12

**Authors:** Masashi Ishikawa, Masae Iwasaki, Hailin Zhao, Junichi Saito, Cong Hu, Qizhe Sun, Atsuhiro Sakamoto, Daqing Ma

**Affiliations:** 1Department of Anesthesiology and Pain Medicine, Graduate School of Medicine, Nippon Medical School, Tokyo 113-8603, Japan; masashi-i@nms.ac.jp (M.I.); masae-a@nms.ac.jp (M.I.); no1-saka@nms.ac.jp (A.S.); 2Division of Anaesthetics, Pain Medicine and Intensive Care, Department of Surgery and Cancer, Faculty of Medicine, Imperial College London, Chelsea & Westminster Hospital, London SW10 9NH, UK; hailin.zhao06@imperial.ac.uk (H.Z.); saitoj@hirosaki-u.ac.jp (J.S.); c.hu15@imperial.ac.uk (C.H.); q.sun17@imperial.ac.uk (Q.S.); 3Department of Anesthesiology, Graduate School of Medicine, Hirosaki University, Hirosaki, Aomori 036-8562, Japan

**Keywords:** microRNA, sevoflurane, desflurane, hypoxia inducible factor-1α, ovarian cancer

## Abstract

Inhalational anaesthetics were previously reported to promote ovarian cancer malignancy, but underlying mechanisms remain unclear. The present study aims to investigate the role of sevoflurane- or desflurane-induced microRNA (miRNA) changes on ovarian cancer cell behaviour. The cultured SKOV3 cells were exposed to 3.6% sevoflurane or 10.3% desflurane for 2 h. Expression of miR-138, -210 and -335 was determined with qRT-PCR. Cell proliferation and migration were assessed with wound healing assay, Ki67 staining and Cell Counting Kit-8 (CCK8) assay with or without mimic miR-138/-210 transfections. The miRNA downstream effector, hypoxia inducible factor-1α (HIF-1α), was also analysed with immunofluorescent staining. Sevoflurane or desflurane exposure to cancer cells enhanced their proliferation and migration. miR-138 expression was suppressed by both sevoflurane and desflurane, while miR-210 expression was suppressed only by sevoflurane. miR-335 expression was not changed by either sevoflurane or desflurane exposure. The administration of mimic miR-138 or -210 reduced the promoting effects of sevoflurane and desflurane on cancer cell proliferation and migration, in line with the HIF-1α expression changes. These data indicated that inhalational agents sevoflurane and desflurane enhanced ovarian cancer cell malignancy via miRNA deactivation and HIF-1α. The translational value of this work needs further study.

## 1. Introduction

Ovarian cancer is the eighth most common cancer among women patients with poor prognosis [1,2]. The 10 year progression-free survival of ovarian cancer remains at 15%, although surgery and related treatment have improved [3]. Epithelial ovarian cancer consists of 90% of ovarian cancer cases [4]. Surgical removal of cancer mass is the first-line therapy for the solid primary cancer including ovarian cancer. Postoperative cancer reoccurrence is one of the key factors leading to poor survival [5], which has been reported to be related to many risk factors including surgical trauma/stress [6] and anaesthetic use [7]. Clinical data suggested that inhalational anaesthetics may affect cancer progression and worsen long-term outcomes in patients with cancer surgery [8,9,10,11]. Our previous study showed that sevoflurane and desflurane exposure increased cell proliferation and migration via metastasis-related gene changes in vitro in an ovarian cancer cell line [12]. In contrast, intravenous anaesthetics including propofol and midazolam were reported to inhibit cancer cell growth [13,14]. Isoflurane enhanced cell proliferation and migration via hypoxia inducible factor-1α (HIF-1α) and matrix metalloproteinase 9 (MMP9) in prostate [13] and ovarian cancer cells [12]. It is well known that HIF-1α has an essential role in cancer cell proliferation and invasion [15,16] whilst MMP9 is crucial in tumour migration and progression [17]. 

MicroRNAs (miRNAs) are noncoding short nucleotides that affect cancer cell biology via protein post-transcription. miRNAs control the normal cell and cancer cell biology [18]. Interestingly, miR-138 [19] and -210 [20] regulate HIF genes, while miR-335 was reported to control MMP9 in glioma [21] and has been suggested to play a role in breast cancer progression [22,23]. However, the potential role of these miRNAs on ovarian cancer development and progression is limited and, furthermore, it remains largely unknown whether inhalational anaesthetics can affect ovarian cancer cell malignancy via miRNA expression changes. The present study, therefore, aims to investigate the potential role of the miRNAs miR-138, -210 and -335 and their downstream effector, HIF-1α and MMP9, as well as changes induced by sevoflurane or desflurane, which are commonly used clinically on ovarian cancer cell biology and malignancy.

## 2. Results

### 2.1. Sevoflurane and Desflurane Increased SKOV3 Cell Proliferation and Migration

#### 2.1.1. Cell Migration

First, the changes in cell proliferation and migration after anaesthetic exposure were evaluated with a wound healing assay, Cell Counting Kit-8 (CCK8) assay and Ki67 immunofluorescent staining. Sevoflurane or desflurane exposure exhibited procancer effects in SKOV3 cells. In the wound healing assay, sevoflurane and desflurane significantly increased the “wound” gap closure ratio at 4 h (sevoflurane 16.82 ± 1.01, *p* < 0.001; desflurane 14.68 ± 1.61, *p* = 0.036 vs. control 12.34 ± 1.68) and at 8 h (sevoflurane 43.59 ± 2.09, *p* < 0.001; desflurane 43.01 ± 1.30, *p* < 0.001 vs. control 34.36 ± 1.86) after gas exposure compared to the control group. There were no differences between sevoflurane and desflurane (*n* = 6) (Figure 1a,b).

#### 2.1.2. Cell Proliferation

Cell proliferation at 24 h after gas exposure was promoted by both anaesthetic agents in the CCK8 assay (sevoflurane 1.07 ± 0.03, *p* = 0.012; desflurane 1.11 ± 0.04, *p* < 0.001 vs. control 1.00 ± 0.04) after gas exposure compared to the control group. There were no differences between sevoflurane and desflurane (*n* = 6) (Figure 1c).

Ki67 immunofluorescence staining showed that cell proliferation after sevoflurane and desflurane exposure significantly increased (sevoflurane 38.22 ± 1.01, *p* < 0.001; desflurane 42.69 ± 1.35, *p* < 0.001 vs. control 28.08 ± 1.52). There was a significant difference between sevoflurane and desflurane (*p* < 0.001, *n* = 6) (Figure 1d,e).

### 2.2. Sevoflurane Downregulated miR-138 and -210 Expression Whereas Desflurane Downregulated Only miR-138 Expression in SKOV3 Cells

#### miRNA Changes after Anaesthesia

To find the miRNAs responsible for the viability changes after anaesthetic exposure, miR-138, -210 and -335 were selected for further investigation. Both sevoflurane and desflurane significantly suppressed miR-138 expression in SKOV3 cells (sevoflurane 0.67 ± 0.21, *p* = 0.020; desflurane 0.61 ± 0.16, *p* = 0.006 vs. control 1.00 ± 0.18, *n* = 6, Figure 1f). Sevoflurane exposure significantly decreased miR-210 expression compared to the other groups (sevoflurane 0.54 ± 0.24, *p* = 0.009; desflurane 0.89 ± 0.22, *p* = 0.686 vs. control 1.00 ± 0.22). There was a significant difference between sevoflurane and desflurane (*p* = 0.046, *n* = 6, Figure 1g) and miR-335 expression was unchanged after any anaesthetic exposure (Figure 1h).

### 2.3. The Mimic of miR-138 and -210 Inhibited Cell Proliferation and Migration after Sevoflurane and Desflurane Exposure

#### 2.3.1. Cell Migration Ability after the Mimic Administration of miR-138 and -210 

To ascertain the involvement of miR-138 and -210 to cell biology changes, the mimics of these miRNAs were transfected to SKOV3 cells before anaesthetic exposure. miR-210 mimic treatment decreased “wound” gap closure ratio at 4 h and 8 h after air exposure as shown in Figure 2a,b (4 h after the exposure: control + miR-210 mimic: 8.90 ± 0.89, *p* = 0.008 vs. control: 12.42 ± 1.47; 8 h after the exposure: control + miR-210 mimic: 23.91 ± 2.51, *p* < 0.001 vs. control: 35.99 ± 1.65, *n* = 6), and miR-138 mimic treatment decreased at 8 h after the exposure (control + miR-138 mimic: 31.04 ± 2.62, *p* = 0.003 vs. control: 35.99 ± 1.65, *n* = 6).

#### 2.3.2. Cell Migration Ability after the Mimic Administration of miR-138 and -210 with Sevoflurane Exposure

Under sevoflurane exposure, the same tendency was observed as summarized in Figure 2c,d; miR-210 mimic treatment decreased “wound” gap closure ratio at 4 h and 8 h after the exposure (4 h after the exposure: sevoflurane + miR-210 mimic 11.91 ± 2.16, *p* = 0.010 vs. sevoflurane 15.75 ± 1.31; 8 h after the exposure: sevoflurane + miR-210 mimic 30.66 ± 3.21, *p* < 0.001 vs. sevoflurane 46.28 ± 1.74, *n* = 6), and miR-138 mimic treatment decreased at 8 h after the exposure (sevoflurane + miR-138 mimic: 38.03 ± 2.42, *p* < 0.001 vs. sevoflurane: 46.28 ± 1.74, *n* = 6). 

#### 2.3.3. Cell Migration Ability after the Mimic Administration of miR-138 and -210 with Desflurane Exposure

With desflurane exposure, only miR-138 mimic treatment suppressed cell migration at 8 h after the exposure (desflurane + miR-138 mimic 38.03 ± 2.42, *p* < 0.001 vs. desflurane 46.03 ± 2.63, *n* = 6, Figure 2e,f). 

#### 2.3.4. Cell Proliferation after the Mimic Administration of miR-138 and -210 with Anaesthetic Exposure

Considering cell proliferation analysis with the CCK8 assay at 2 h after the exposure, miR-138 mimic treatment reduced cell proliferation at any anaesthetic exposure, but miR-210 mimic treatment decreased under control and sevoflurane exposure, not desflurane exposure (*n* = 6, Figure 2g,h,i). 

### 2.4. Sevoflurane and Desflurane Exposure Enhanced HIF-1α Protein Expression Which Was Reverted by miR-138 and -210 Mimic Treatment

#### 2.4.1. Immunofluorescent Staining Analysis

miR-138 and -210 are reported to control HIF-1α expression, a key factor of cancer malignancy. As shown in Figure 3a,b, HIF-1α expression in cytoplasm elevated after sevoflurane and desflurane exposure (sevoflurane 1.57 ± 0.23, *p* < 0.001, desflurane 1.29 ± 0.14, *p* = 0.019, vs. control 1.00 ± 0.08, *n* = 6), which significantly lessened with miR-138 and -210 mimic treatment at any anaesthetic condition (control 1.00 ± 0.08, control + miR-210 mimic 0.77 ± 0.12, *p* = 0.003, control + miR-138 mimic 0.85 ± 0.08, *p* = 0.065, vs. control 1.00 ± 0.08; sevoflurane + miR-210 mimic 0.58 ± 0.14, *p* < 0.001, sevoflurane + miR-138 mimic 0.59 ± 0.14, *p* < 0.001, vs. sevoflurane 1.57 ± 0.23; desflurane + miR-210 mimic 0.77 ± 0.19, *p* < 0.001, desflurane + miR-138 mimic 0.78 ± 0.10, *p* < 0.001, vs. desflurane 1.29 ± 0.14, *n* = 6). 

#### 2.4.2. Western Blotting Analysis

The Western blotting analysis showed similar trend changes but did not reach any statistical significance (control + miR-210 mimic 0.92 ± 0.11, *p* = 0.905, control + miR-138 mimic 0.78 ± 0.26, *p* = 0.231 vs. control 1.00 ± 0.15; sevoflurane + miR-210 mimic 1.07 ± 0.13, *p* = 0.961, sevoflurane + miR-138 mimic 0.87 ± 0.33, *p* = 0.103 vs. sevoflurane 1.11 ± 0.12; desflurane + miR-210 mimic 1.07 ± 0.09, *p* = 0.936, desflurane + miR-138 mimic 0.81 ± 0.28, *p* = 0.224 vs. desflurane 1.01 ± 0.18, *n* = 6–7, Figure 3c,d).

## 3. Discussion

The present study with an ovarian cancer cell line (SKOV3) demonstrated that inhalational anaesthetic agents exert procancer effects via miR-138 and/or miR-210/HIF-1α modulations. Furthermore, miR-210 and -138 mimic administration attenuated the protumour effects of inhalational anaesthetics with HIF-1α suppression. Taken together, both sevoflurane and desflurane may promote cancer cell migration and proliferation by direct HIF-1α upregulation via miR-138 and -210 suppression.

Our data demonstrated that sevoflurane and desflurane, commonly used for surgical anaesthesia, may contribute to the risk of cancer recurrence after surgery that may be involved with the HIF-1α signalling pathway via miRNA mediation (Figure 4). The importance of miRNAs in cancer cell biology is well documented in the literature. miRNAs can be pro- or anticancer regulators due to their complex functions because a single miRNA can regulate more than 100 target genes and proteins [24]. miRNAs were reported to regulate cancer cell proliferation and invasion in ovarian cancer cells [25]. Indeed, miR-630 promoted SKOV3 cell proliferation and migration by targeting Krüppel-like factor 6 (KLF6) [26]. miR-18b accelerated invasion and migration of ovarian cancer cells via phosphatase and tensin homolog (PTEN) [27]. In contrast, miR-34 reduced the mesenchymal-to-epithelial transition (MET) protein levels and suppressed cancer cell migration, invasion and proliferation in p53-null SKOV3 cells [28]. miR-335 is reported to control MMP9 in glioma [21], as suggested in breast cancer [22,23]. Furthermore, miRNA changes after inhalational anaesthetic exposure can modulate cancer cell biology. For example, sevoflurane can inhibit cell migration and invasion in colon cancer cells via the ERK/MMP9 pathway by regulating miR-203 [29]. On the other hand, desflurane enhanced colorectal cancer malignancy via the miR-34a/lysyl oxidase-like 3 axis [30]. Our data clearly demonstrated that sevoflurane and desflurane both promoted SKOV3 cancer cells, one of the ovarian cancer types, and malignancy via miR-138 and/or -210. The discrepancy was likely due to the differences of experimental conditions; the concentrations of sevoflurane or desflurane were exposed to cancer cells for 6 h in those previous studies [29,30], which caused cancer cell death due to their toxicity of cumulative doses. It is worth pointing out that those previous studies were designed beyond clinical settings as anaesthesia was administered for 6 h and this is exceedingly rare for cancer surgery clinically. Therefore, the translational value of those studies is very questionable. 

Several studies showed that HIF-1α was directly regulated by miRNAs including miR-138 [19] and miR-210 [20]. Our data showed that sevoflurane and desflurane downregulated the expression of miR-138 in SKOV3 cells. miR-138 is a tumour suppressor in several types of cancer cells [31,32,33,34]. miR-138 expression is downregulated in numerous cancers including glioblastoma [35], and its downregulation can suppress cell proliferation, invasion and migration as a tumour-suppressor gene via targeting of Histone’s H2A variant (H2AX) in cervical cancer cells [36] or via Sry-Related HMG-BOX-4 (SOX4) in ovarian cancer cells [35]. The expression level of miR-138 correlates with ovarian cancer prognosis and, indeed, the level of miR-138 in patients with lymphatic metastasis was lower than that in patients without lymphatic metastasis [32]. Patients with low miR-138 expression tend to be in late stage and have malignant phenotypes [35]. miR-138 inhibits the occurrence and development of ovarian cancer by downregulating the expression of SOX12 gene [36] and invasion via SOX4 and HIF-1α oncogenic transcriptional factors [37]. HIF-1α is one of the targets of miR-138 and miR-138 negatively regulates HIF-1α to suppress proliferation, invasion and migration in melanoma [19,38] and ovarian cancer [37]. miR-210 is not only a tumour-stimulating miRNA but has a dual role as a procancer and anticancer miRNA. miR-210 can act as a tumour suppressor, inhibiting cell proliferation in ovarian cancer [39] and laryngeal squamous cell carcinoma [40]. miR-210 can attenuate cancer cell activity through downregulating E2F3, fibroblast growth factor receptor-like 1, homeobox protein Hox-A1, homeobox protein Hox-A9 and Max-binding protein [41,42,43]. HIF-1α contains a miR-210 targeting site in its 3′ UTR and is one of miR-210’s target genes [20]. Unlike desflurane, we found that sevoflurane also decreased miR-210 expression in this study but both agents were ineffective on miRNA-355. The underlying mechanisms of these differences remain unknown and warrant further study.

Inhalational anaesthetic isoflurane was reported to increase ovarian cancer cell proliferation and migration [44], and all commonly used inhalational anaesthetics can even alter cancer-metastasis-related genes [12]. All these may influence the ovarian cancer surgical outcomes, although it was shown that desflurane was associated with a lower recurrence rate compared with sevoflurane, but underlying mechanisms for such differences remain unknown [45]. Previous studies demonstrated that inhalational anaesthetics could be procancer factors in other cancer types. Isoflurane exposure to prostate cancer cells promoted HIF-1a and its downstream effector expressions including vascular endothelial growth factor (VEGF), and it increased HIF-1α expression in a concentration- and time-dependent manner; HIF-1α was translocated from the cytoplasm to the nucleus as a transcriptional factor, resulting in promoted proliferation and invasion [13]. In ovarian cancer cells, CXC chemokine receptor 2 (CXCR2), VEGF-A, MMP11 and transforming growth factor β (TGF-β) expressions were all significantly increased by inhalational anaesthetics, indicating the activation of key molecular mediators of cancer cell proliferation, cell migration and angiogenesis [12]. HIF-1α has been identified as one of the key regulators in tumour progression, cancer cell proliferation, invasion and angiogenesis [46,47] by the activation of the phosphoinositide 3-kinase/Akt/mammalian target of rapamycin (PI3K/Akt/mTOR) and mitogen-activated protein kinase/extracellular signal-regulated kinase (MAPK/ERK) pathways [48]. The increased HIF-1α expression has been found in numerous cancer cells [49] and clinically linked with tumour growth, metastasis and poor clinical prognosis [50]. HIF-1α also regulates the transcription of multi-specific drug efflux transporters leading to chemoresistance [51].

The present study was not free from limitations. Firstly, this is an in vitro study with one cell line of ovarian cancer types. Furthermore, SKOV3 unlikely represents the most common ovarian cancer cell type of high-grade serous carcinoma (HGSC) [52,53] and each of the ovarian cancer cell phenotypes have their own biological characteristics. Thus, further investigations using other cell lines of ovarian cancers are needed to see whether the current findings are also evident in other ovarian cancer cells types. If so, then the onco-effects of inhalational anaesthetics are also needed to be verified in a clinic scenario mode and an in vivo setting. Secondly, the wound healing assay was used in our study; it measures both cell migration and proliferation. Therefore, the separate effect of anaesthetics studied on migration or proliferation warrants further study. Thirdly, the mechanisms as to why sevoflurane exposure suppressed miR-138 and -210 expression while desflurane only suppressed miR-138 expression (Figure 1) remain unknown. Whether this discrepancy is due to their very different chemical structure, physical properties and anaesthetic potency is elusive. Interestingly, this is in line with our previous findings showing metastatic gene expressions induced by them in SKOV3 were also different [12]. Furthermore, we did not evaluate the HIF-1α downstream effectors and, hence, their effects on cancer cell biology are unknown. Lastly, it is worth pointing out that protein expression analysis with miRNA-mimic administration under anaesthetic exposure was not conclusive when compared to immunostaining (Figure 3). It was very likely that there were considerably more experimental procedures involved in Western blot analyses which may distort the results whilst, arguably, it may also mean that whether in situ immunostaining is more reliable than Western blot is an open question.

In summary, our data indicated that sevoflurane or desflurane promoted SKOV3 cell migration and proliferation via miR-138 and -210 suppression. Although the clinical significance of our work requires further investigation, this study undoubtedly enhances our understanding of the impact of inhalational anaesthesia on cancer cells during the perioperative period and provides supporting evidence for the selection of appropriate anaesthetics for the benefit of patients’ prognosis after surgery. 

## 4. Materials and Methods

### 4.1. Cell Culture

SKOV3 human ovarian epithelial carcinoma cell line (European Cell Culture Collection, Salisbury, UK) was cultured at 37 °C in a humidified atmosphere containing 5% CO_2_ balanced with air in McCoy’s 5A medium (Sigma-Aldrich, Dorset, UK) supplemented with 10% fetal bovine serum (Thermo Scientific, Epsom, UK), 2 mM L-glutamine (Sigma-Aldrich) and 1% penicillin (Sigma-Aldrich) for the experiments described below with or without inhalational agent exposure for further analyses. 

### 4.2. Inhalational Anaesthetic Exposure

When cultures reached 60%, they were exposed to 21% O_2_, 5% CO_2_ and 3.6% sevoflurane or 10.3% desflurane balanced with N_2_ (BOC, South Humberside, UK) in a purpose-built 1.5 L airtight gas chamber, equipped with inlet and outlet valves. The chamber was then placed in an incubator (Galaxy R CO_2_ chamber; New Brunswick Scientific, Enfield, CT, USA) at 37 °C for 2 h. Other cohort cells were exposed to the same concentration gases without any inhalational anaesthetics and served as controls. After exposure, cells were returned to the normal culture incubator for further study. Both sevoflurane and desflurane were set at the equipotent concentrations approximately equal to 1.7 minimum alveolar concentrations in human. 

### 4.3. Wound Healing Assay

Cells (2.5 × 10^4^) were seeded into each well (Culture-Insert 3 wells; Ibidi, Martinsried, Germany), incubated for 24 h before wound was made and then exposed to the experimental gas mixtures as above for 2 h. The gap closure was monitored under the microscope and a digital camera (CK30-SLP; Olympus, Tokyo, Japan) at 0, 4 and 8 h after gas exposure. Images were analysed using Image J version 1.52a software (National Institute of Health, Bethesda, MD, USA). 

### 4.4. Cell Proliferation Test

Cells (7 × 10^3^) were seeded in 96-well plates and exposed to the experimental gas mixture. Cell proliferation was assessed by the CCK8 reagent (Sigma-Aldrich) which was read from culture media at 450 nm (ELx800 Microplate Reader, BioTek Instruments, Winooski, VT, USA) at 24 h after gas exposure. Cell proliferation was expressed relative to the corresponding control.

### 4.5. RNA Extraction and Reverse Transcription

Immediately after the anaesthetic exposure, total RNA was extracted from cells using QIAzol Lysis Reagent (Qiagen, West Sussex, UK) and an miRNeasy Mini Kit (Qiagen) following the manufacturer’s protocols. RNA quantity and quality were assessed using a BioPhotometer (Eppendorf, Stevenage, UK). Samples with an A260/A280 ratio > 1.8 were considered to be sufficient quality for further analysis. A total of 1 ng RNA was converted to cDNA using miScript II RT Kit (Qiagen) with the thermal cycles of 37 °C for 60 min and 95°C for 5 min (Mastercycler^®^, Eppendolf, Stevenage, UK).

### 4.6. qRT-PCR

qRT-PCR was performed with the miScript SYBR Green PCR Kit (Qiagen) and Rotor gene Q system (Qiagen). SNORD44 small nuclear RNA was used as an endogenous control. The specific primers, miR-138-5p, miR-210-3p and miR-335-5p, were purchased from Qiagen. Thermal cycles were as follows: 95 °C for 15 min and 40 cycles of 94 °C for 15 s, 55 °C for 30 s and 70 °C for 30 s. Melting curve analysis was used to confirm the specificity of amplification. The relative expressions of miRNAs to SNORD44 were determined using the comparative 2^-ΔΔCt^ method. A 2-fold change or over was considered to be of significance.

### 4.7. miRNA Mimic Transfection

The mimics of miR-138-5p (5′AGCUGGUGUUGUGAAUCAGGCCG) and -210 (5′CUGUGCGUGUGACAGCGGCUGA) were purchased from Qiagen in comparison with All Stars Negative Control siRNA (Qiagen) as the negative control. Cells were transfected with either 50 nM of the mimic or the negative control using the HiPerFect transfection reagent (Qiagen), following the manufacturer’s protocols. After 24 h of transfection, the transfection solution was replaced with fresh medium before the experimental gas exposure.

### 4.8. Immunofluorescent Staining

Cells at 24 h after gas exposure were fixed in 4% paraformaldehyde for 10 min and blocked with 10% normal donkey serum (Sigma-Aldrich) for 1 h, followed by overnight incubation at 4 °C in primary antibodies: rabbit polyclonal anti-Ki-67 antibody (1:500; Abcam PLC, Cambridge, UK) and rabbit polyclonal anti-HIF-1α antibody (Novus Biologicals, Oxford, UK). Cells were incubated in Alexa flour 568-conjugated secondary antibody (ThermoFisher scientific) on the following day. Then, cells were costained for the cell nuclei with Vectashield mounting medium containing nuclear dye 4′,6-diamidino-2-phenylindole–mounting medium (Millipore). Slides were then examined using a BX60 wide-field fluorescence microscope (Olympus, Hamburg, Germany) and AxioCam camera (Zeiss, Oberkochen, Germany) with Zeiss software under 20× magnification. Images were captured using a camera. Fluorescence intensity was quantified as the mean pixel intensity of relevant antibody staining using Image J (version 1.52a, National Institute of Health). Ten representative regions per slide were randomly selected. Intensity values were calculated and expressed as relative to the control.

### 4.9. Western Blotting

The protein was extracted from cell samples using a cell lysis buffer (Cell Signalling Technology, Hitchin, UK) at 24 h after gas exposure and quantified with a Bradford protein assay (Bio-Rad Laboratories, Hercules, CA, USA). Then, 60 µg of protein of each sample was loaded into NuPAGE^®^ 4–12% Bis-Tris Precast Gels (Thermo Scientific) for electrophoresis. After electrophoresis, proteins were transferred onto a polyvinylidenedifluoride membrane using the iBlot^®^2 Dry Blotting System (Thermo Scientific). Membranes were blocked with 5% nonfat powdered milk in Tris-buffered saline with Tween for 1 h at room temperature, then incubated overnight at 4 °C with rabbit anti-HIF-1α primary antibody (1:500; Abcam) followed by horseradish peroxidase-linked antirabbit secondary antibody (1:1000; Cell Signalling Technology) for 1 h. Protein bands were visualised using the Enhanced Chemiluminescence system (Santa Cruz, Dallas, TX, USA) and Syngene GeneSnap software (Syngene, Cambridge, UK). The intensity of grey-scale protein bands was assessed using ImageJ software.

### 4.10. Statistical Analysis

All numerical data are presented as dot plots and expressed as mean ± SD. One-way analysis of variance followed by post-hoc Tukey’s test were used for data analyses with Prism version 8.0 (GraphPad Software, San Diego, CA, USA). A *p* value less than 0.05 was considered to be a statistical significance.

## 5. Conclusions

In conclusion, our data demonstrated that inhalational anaesthetics enhanced some ovarian cancer cell biology via miRNA downregulation. The regulatory mechanisms of HIF-1α expression via miRNAs may contribute to the detrimental effects of inhalational anaesthetics and hence worsen cancer outcomes. This research is a proof-of-concept study with one cell line. In order to investigate “common and general onco-effects of inhalational agents”, more cell lines need to be considered in future studies. These data may warrant more research to refine optimal anaesthetic regimens for cancer patients including ovarian cancer sufferers.

## Figures and Tables

**Figure 1 ijms-22-01826-f001:**
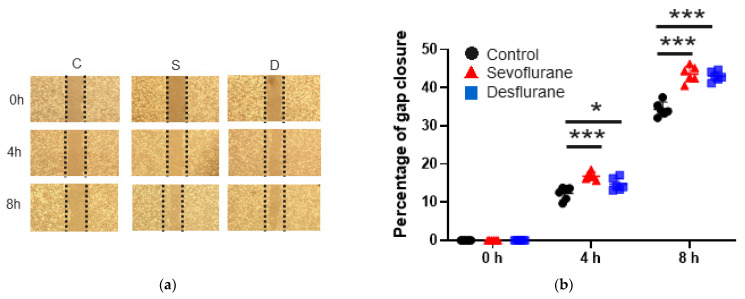
The changes of cell viability and miRNAs after inhalational anaesthesia. (**a**) SKOV3 cell migration analysis with wound healing assay after 2 h of inhalational anaesthesia: control (left), 3.6% sevoflurane (middle) and 10.4% desflurane (right), at 0 h later (upper), 24 h later (middle) and 48 h later (bottom). The microscopic images at 0, 24 and 48 h after general anaesthesia. (**b**) The comparison between anaesthetics in percentage of gap closure by wound healing assay. (**c**) Cell proliferation analysis with CCK8 assay relative to control group. (**d**) Ki67 immunofluorescence staining: Ki67 (red), marker for cell proliferation, in control (left), sevoflurane- (middle) and desflurane-treated (right) SKOV3 cells, counterstained with DAPI (blue); ×20 magnification, scale bar = 20 μm. (**e**) Comparison of percentage of Ki67 positive cells at 24 h after anaesthesia exposure. **(f**–**h)** miRNA expressions evaluated with qRT-PCR compared to control group just after anaesthesia exposure: (**f**) miR-138 (HIF-1α regulator), (**g**) miR-210 (HIF-1α regulator) and (**h**) miR-335 (MMP9 regulator). Data showed as plots and mean ± SD. * *p* < 0.05, ** *p* < 0.01, *** *p* < 0.001, *n* = 6, one-way ANOVA with Tukey-Kramer compared to the control group. C: control, S: sevoflurane, D: desflurane and CCK8: cell count kit 8.

**Figure 2 ijms-22-01826-f002:**
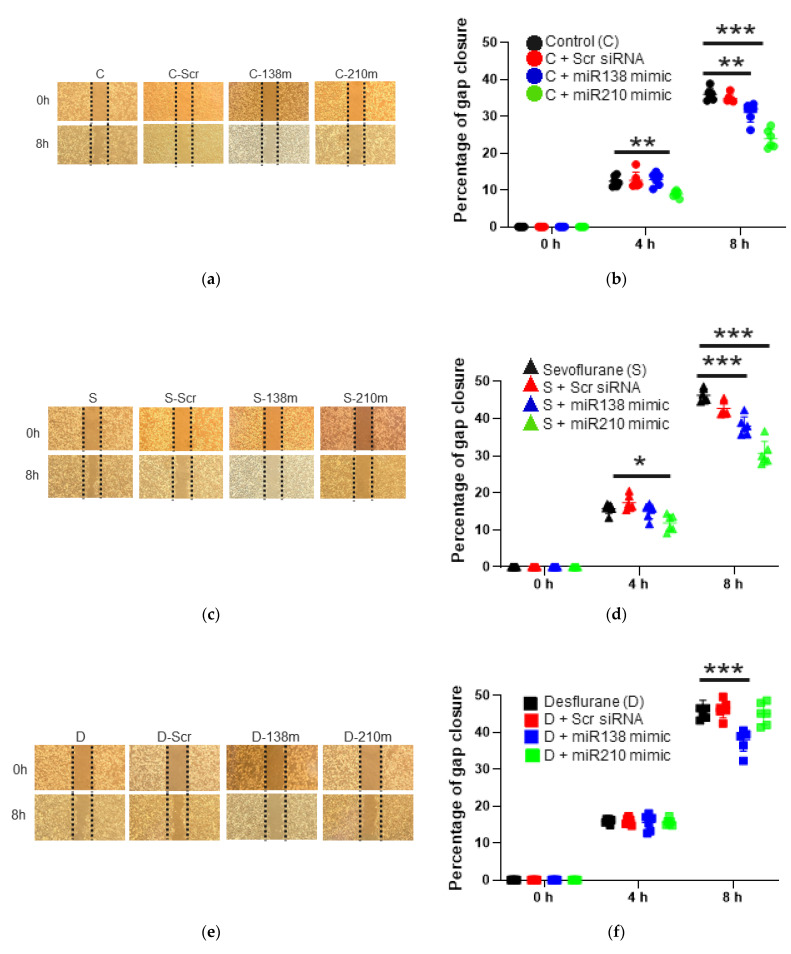
The changes of cell migration and proliferation after inhalational anaesthesia with mimic miRNA pretreatment. (**a**–**f**) SKOV3 cell migration analysis with wound healing assay after 2 h of inhalational anaesthesia with miRNA inhibition pretreatment at 0 h (upper) and 8 h (bottom) after anaesthesia. (**a**) The microscopic images after control anaesthesia. (**b**) The comparison of gap closure percentage with control anaesthesia and mimic miRNA pretreatment. (**c**) The microscopic images after 3.6% sevoflurane anaesthesia with mimic miRNA pretreatment. (**d**) The comparison of gap closure percentage with 3.6% sevoflurane anaesthesia and mimic miRNA pretreatment. (**e**) The microscopic images after 10.4% desflurane anaesthesia with mimic miRNA pretreatment. (**f**) The comparison of gap closure percentage with 10.4% desflurane anaesthesia and mimic miRNA pretreatment. (**g**–**i**) Cell proliferation analysis with CCK8 assay with anaesthesia and mimic miRNA pretreatment relative to each control group at 24 h after exposure: (**g**) control anaesthesia, (**h**) 3.6% sevoflurane anaesthesia and (**i**) 10.4% desflurane anaesthesia. Data showed as plots and mean ± SD. * *p* < 0.05, ** *p* < 0.01, *** *p* < 0.001, *n* = 6. One-way ANOVA with Tukey-Kramer compared to each control group. C: control, S: sevoflurane, D: desflurane, CCK8: Cell Counting Kit-8, Scr: scrambled miRNA, 138m: miR-138 mimic and 210m: miR-210 mimic.

**Figure 3 ijms-22-01826-f003:**
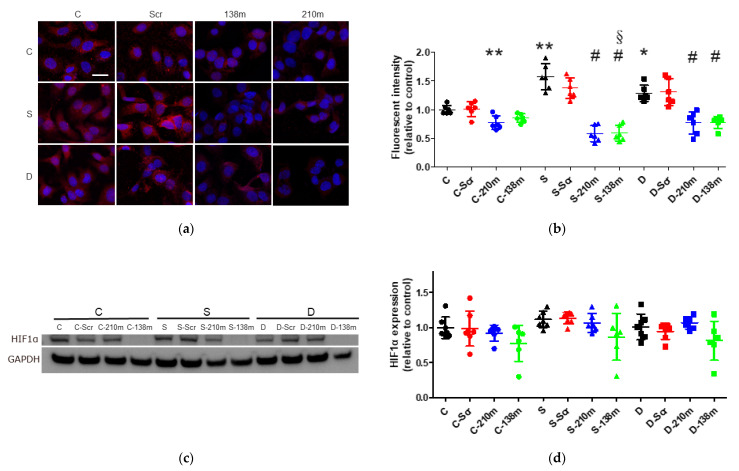
The HIF-1α expression changes after inhalational anaesthesia with mimic miRNA pretreatment. (**a**) HIF-1α immunofluorescence staining, HIF-1α (red) in control (left), sevoflurane- (middle) and desflurane-treated (right) SKOV3 cells with mimic miRNA pretreatment, counterstained with DAPI (blue); ×20 magnification, scale bar = 20 μm. (**b**) The comparison of the HIF-1α immunofluorescence intensity with each anaesthetic and mimic miRNA pretreatment. (**c**) The representative image of HIF-1α Western blotting analysis with anaesthesia and mimic miRNA pretreatment. (**d**) The comparison of HIF-1α expression in Western blotting after anaesthesia and mimic miRNA pretreatment. Data showed as plots and mean ± SD. * *p* < 0.05, ** *p* < 0.01, *n* = 6. One-way ANOVA with Tukey-Kramer compared to the control group, # *p* < 0.05; one-way ANOVA with Tukey-Kramer compared to each control group, § *p* < 0.05; one-way ANOVA with Tukey-Kramer compared to the control group with the same mimic miRNA pretreatment. C: control, S: sevoflurane, D: desflurane, CCK8: Cell Counting Kit-8, Scr: scrambled miRNA, 138m: miR-138 mimic and 210m: miR-210 mimic.

**Figure 4 ijms-22-01826-f004:**
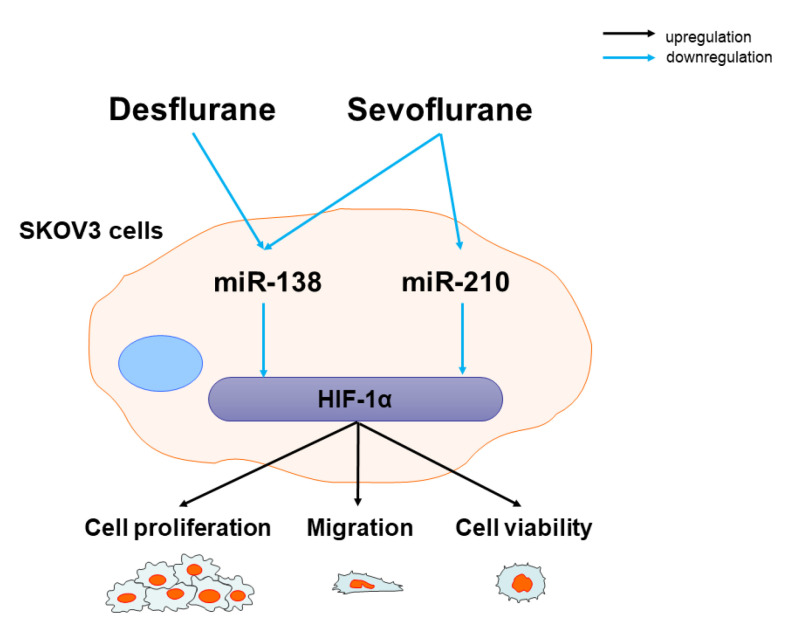
Inhalational anaesthetics enhanced SKOV3 cell malignancy via downregulation of anticancer miRNA expressions. Sevoflurane and desflurane exposure to SKOV3 cells increase SKOV3 cell proliferation and migration via HIF-1α by miR-138 downregulation. Only sevoflurane decreased miR-210 expression leading to an enhancement of SKOV3 cell malignancy via HIF-1α.

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
