# Peer review of "Sevoflurane and Desflurane Exposure Enhanced Cell Proliferation and Migration in Ovarian Cancer Cells via miR-210 and miR-138 Downregulation"

_ijms, 2021, doi:10.3390/ijms22041826_

Round 1
Reviewer 1 Report
Dear Author,
I read with attention your manuscript entitled “Sevoflurane and desflurane exposure enhanced cell proliferation, migration and viability in ovarian cancer cells via miR-210 and miR-138 upregulation”.
It refers to in vitro experiments aiming at exploring the pro-oncogenic effect of two anesthetic gas commonly used in daily clinical practice on 1 ovarian cancer cell line. The issue is of importance. Despite the fact that we have known for several years the effect of certain anesthetics the data on the underlying mechanism of action remain poor.
Two miRNA and HIFa have been explored as factors responsible for this detrimental effect.
The introduction is concise and clear.
The experimental plan is logical and well presented with adapted tests.
The results are clear.
The discussion appeared balanced to me, however, the hypothesis of another regulatory miRNA explaining the slight difference (not significant most of the time) between miR-138 and -210 could be explored.
My only complaint would be that Figure 1 could probably be improved by adding colors in the graphs (b,c,e, and f). In all the figures the scale could be modified to enhance the visual effect of some of the significant differences.
Very good paper ++
Yours sincerely,
Author Response
Response to Reviewer 1 Comments
Point 1: My only complaint would be that Figure 1 could probably be improved by adding colors in the graphs (b,c,e, and f). In all the figures the scale could be modified to enhance the visual effect of some of the significant differences.
Response 1: Thank you very much for your constructive comments. We have amended Figure 1 accordingly.
Reviewer 2 Report
The authors demonstrated the potential of inhalational anesthetics to ovarian enhance cancer cell progression via miR-138 and miR-210 downregulation. Although their finding is interesting, the rationale for some experiments is not well described. The influence of inhalational anesthetics on the expression of HIF1a is not convincing, either. I believe more careful interpretation is required for the result shown in figure 2 and figure 3.
Major points
The title says sevoflurane and desflurane upregulate miR-210 and miR-138 which are totally opposite to the results.
The authors should provide a logical explanation why they focused on miR-138, miR-210 and miR-335 in this study.
I figure 1, I recommend the authors to additionally perform trans-well assay and to check EMT markers to validate the influence of sevoflurane or desflurane on migration,
In figure 2 and 3, the reason why the authors investigated the influence of miR-138 and miR-210 mimic transfection under sevoflurane or desflurane exposure is unclear. Please clearly describe the purpose of this experimental setting. Interpretation or discussion about the result in figure 2 are missing as well.
The blots shown figure 3-c do not represent the quantified data shown in figure 3-d. It seems that desflurane potentially suppresses HIF1a expression, and miR-138 mimic transfection strongly suppresses HIF1a expression regardless of inhalational anesthetics. Again, the significance of examining influence of miR-138 and miR-210 mimic transfection under sevoflurane or desflurane exposure is unclear.
To validate the association of miR138 and miR210-mediated HIF1a regulation on the phenotypic change shown in figure 1, several signal pathways should be also assessed by western blotting which are downstream of HIF1a and involved in cancer cell proliferation or migration ability
Ovarian cancer is composed of several subtypes. As the character of each subtype is quite distinct from the others, it is now widely recommended that ovarian cancer cells retaining the features of their original cancer subtype should be selected according to the experimental purpose. Whereas high grade serous carcinoma (HGSC) is the most common subtype occupying up to 70 % of ovarian cancer incidence, SkOV3 is documented as an unlikely representative of HGSC.( PMDI 23839242, 25230021). I recommend the authors to validate the reproducibility of the main results in the manuscript with additional cell line(s) representing HGSC as well.
Minor points
About Line 76-80. Ki67 is a marker for cell proliferation rather than cell viability.
Blots and error bars are unclear in figure 1-b. figure 2-b, 2-d, 2-f.
I could not understand the meaning of ",a cell proliferation marker" in line 168.
Author Response
Response to Reviewer 2 Comments
Point 1: The title says sevoflurane and desflurane upregulate miR-210 and miR-138 which are totally opposite to the results.
Response 1: We thank the reviewer for pointing out this. We have amended the title.
Point 2: The authors should provide a logical explanation why they focused on miR-138, miR-210 and miR-335 in this study.
Response 2: Thank you for your excellent comments. We have amended the background and the discussion accordingly(page 2, lines 49-55) (page 7, lines 163-164).
Point 3: In figure 1, I recommend the authors to additionally perform trans-well assay and to check EMT markers to validate the influence of sevoflurane or desflurane on migration,
Response 3: We thank the reviewer for this recommendation. We agree that the transwell assay and EMT markers can be used to evaluate the migration whereas the wound healing assay can reflect both migration and proliferation of cancer cells. We will actively consider these methods in our future studies. We have discussed these in the discussion section (page 9, lines 215-216).
Point 4: In figure 2 and 3, the reason why the authors investigated the influence of miR-138 and miR-210 mimic transfection under sevoflurane or desflurane exposure is unclear. Please clearly describe the purpose of this experimental setting. Interpretation or discussion about the result in figure 2 are missing as well.
Response 4:
This is a great suggestion.
We have amended the result and the discussion sections to clarify these points (page 7, lines 149-154).
Point 5: The blots shown figure 3-c do not represent the quantified data shown in figure 3-d. It seems that desflurane potentially suppresses HIF1a expression, and miR-138 mimic transfection strongly suppresses HIF1a expression regardless of inhalational anesthetics. Again, the significance of examining influence of miR-138 and miR-210 mimic transfection under sevoflurane or desflurane exposure is unclear.
Response 5:
Thank you for this comment. The mimic administration was needed to confirm that sevoflurane and desflurane could directly modulate miRNA expressions. Figure 3 showed the protein expression analysis after mimic administration and anaesthesia exposure. The suppressing effect of mimic administration on HIF1a could be greater than the promoting effect of inhalational anaesthetics, so there were no significant changes due to these counteracting effects. As these data do not add much value for our work and hence conclusions, we decided to omit these data.
Point 6: To validate the association of miR138 and miR210-mediated HIF1a regulation on the phenotypic change shown in figure 1, several signal pathways should be also assessed by western blotting which are downstream of HIF1a and involved in cancer cell proliferation or migration ability
Response 6:
This is a great point. It would be useful to evaluate the downstream proteins under HIF1a for the investigation of the micro-mechanism in cancer cell viability. However, the pivotal roles of HIF1a and its downstream proteins are already widely recognised in vitro and in vivo (PMID 25016511). The main purpose of the present in vitro study was to clarify the direct effect of inhalational anaesthetics on HIF1a via miRNA expression changes. Thus, we did not evaluate the downstream proteins. There could be other mechanisms which could modulate HIF1a or other key markers after anaesthesia, leading to the cell viability changes.
We added these points to the discussion part (page 9, lines 216-217).
Point 7: Ovarian cancer is composed of several subtypes. As the character of each subtype is quite distinct from the others, it is now widely recommended that ovarian cancer cells retaining the features of their original cancer subtype should be selected according to the experimental purpose. Whereas high grade serous carcinoma (HGSC) is the most common subtype occupying up to 70 % of ovarian cancer incidence, SkOV3 is documented as an unlikely representative of HGSC. (PMDI 23839242, 25230021). I recommend the authors to validate the reproducibility of the main results in the manuscript with additional cell line(s) representing HGSC as well.
Response 7: We thank the reviewer for this excellent comment.
We agreed that one subtype of ovarian cancer was used in the present study as one of the limitation, so we have clarified this in the discussion section (page 8, lines 213-214).
Minor points
Point 8: About Line 76-80. Ki67 is a marker for cell proliferation rather than cell viability.
Response 8: Thank you very much for pointing this. We have amended the result section in the manuscript accordingly (page 2, lines 70-72) (page 4, lines 88).
Point 9: Blots and error bars are unclear in figure 1-b. figure 2-b, 2-d, 2-f.
Response 9: Thank you very much for pointing this. We have amended the figures accordingly.
Point 10: I could not understand the meaning of ",a cell proliferation marker"in line 168.
Response 10: We thank the reviewer pointing this. We have amended the manuscript (page 7, lines 140).
Round 2
Reviewer 2 Report
I appreciate the revision effort by the authors. I’m afraid to say that the authors have not respond to all of my concerns at this point.
In figure 2 and figure 3;
The purpose of transfecting miR-mimics under sevoflurane or desflurane exposure is still not appropriately described. The authors mentioned that the purpose is to confirm sevoflurane and desflurane can directly modulate miRNA expression only in their reply, but I do not believe this experimental setting does not provide any evidence of direct modulation. It simply seems a rescue assay to confirm that the observed phenotypic change was led by miR138 or miR210 downregulation . Again, please add a comprehensive explanation about the purpose of these experiments. It might be also informative to add a focused discussion about the observed difference between sevoflurane and desflurane exposure with miR-210 mimic transfection in association with the findings in figure 1.
In figure 3;
I need to point out that the omitted figures (previous figure 3-c and d) does influence the conclusion the authors made. If the authors believe immunofluorescent assay is more reliable than WB, it is preferable to add discussion on it rather than eliminating the conflicting result from the point of research ethics.
In discussion;
As well indicated in plenty of studies, SkOV3 is not a representative of the majority of ovarian cancer even though it is one of the most frequently used cells in ovarian cancer research. Conclusions led by results with SkOV3 alone are not convincing as long as a study is focusing on ovarian cancer in general. If the authors decided not to try other cells in the current project, please at least add much more careful discussion about this point with enough references.
Author Response
Response to Reviewer 2 Comments
Point 1: In figure 2 and figure 3;
The purpose of transfecting miR-mimics under sevoflurane or desflurane exposure is still not appropriately described. The authors mentioned that the purpose is to confirm sevoflurane and desflurane can directly modulate miRNA expression only in their reply, but I do not believe this experimental setting does not provide any evidence of direct modulation. It simply seems a rescue assay to confirm that the observed phenotypic change was led by miR138 or miR210 downregulation. Again, please add a comprehensive explanation about the purpose of these experiments.
Response 1:
Thanks! In order to clarify the onco-effects of anesthetics via miRNA downregulation, MiRNA mimics were used to verify this association. The similar method with mimic/inhibitor application has been well established (PMID: 24950164, 31788099). We have discussed this (Page 8, line 174-175)
Point 2: It might be also informative to add a focused discussion about the observed difference between sevoflurane and desflurane exposure with miR-210 mimic transfection in association with the findings in figure 1.
Response 2:
This is a great suggestion. We’ve discussed the difference of anesthetics, Sevoflurane and Desflurane, in the discussion Section (page 9, line 227-230).
Point 3: In figure 3;
I need to point out that the omitted figures (previous figure 3-c and d) does influence the conclusion the authors made. If the authors believe immunofluorescent assay is more reliable than WB, it is preferable to add discussion on it rather than eliminating the conflicting result from the point of research ethics.
Response 3:
Thank you for your suggestion. We added WB results and discussion on the conflicting result (page 9, line 232-236).
Point 4: In discussion;
As well indicated in plenty of studies, SkOV3 is not a representative of the majority of ovarian cancer even though it is one of the most frequently used cells in ovarian cancer research. Conclusions led by results with SkOV3 alone are not convincing as long as a study is focusing on ovarian cancer in general. If the authors decided not to try other cells in the current project, please at least add much more careful discussion about this point with enough references.
Response 4:
Thank you very much for pointing out this. We’ve amended this point and added sentences in the discussion and conclusion section (page 9, line 220-225).